# Observation of higher-order non-Hermitian skin effect

Xiujuan Zhang[1], Yuan Tian[1], Jian-Hua Jiang [2✉], Ming-Hui Lu [1,3✉] & Yan-Feng Chen [1,3✉]

Beyond the scope of Hermitian physics, non-Hermiticity fundamentally changes the topological band theory, leading to interesting phenomena, e.g., non-Hermitian skin effect, as confirmed in one-dimensional systems. However, in higher dimensions, these effects remain elusive. Here, we demonstrate the spin-polarized, higher-order non-Hermitian skin effect in two-dimensional acoustic higher-order topological insulators. We find that non-Hermiticity drives wave localizations toward opposite edges upon different spin polarizations. More interestingly, for finite systems with both edges and corners, the higher-order non-Hermitian skin effect leads to wave localizations toward two opposite corners for all the bulk, edge and corner states in a spin-dependent manner. We further show that such a skin effect enables rich wave manipulation by configuring the non-Hermiticity. Our study reveals the intriguing interplay between higher-order topology and non-Hermiticity, which is further enriched by the pseudospin degree of freedom, unveiling a horizon in the study of non-Hermitian physics.

[1] National Laboratory of Solid State Microstructures and Department of Materials Science and Engineering, Nanjing University, Nanjing, China. [2] School of Physical Science and Technology, and Collaborative Innovation Center of Suzhou Nano Science and Technology, Soochow University, Suzhou, China. [3] Collaborative Innovation Center of Advanced Microstructures, Nanjing University, Nanjing, China. ✉email: joejhjiang@hotmail.com; luminghui@nju.edu.cn; yfchen@nju.edu.cn

Hermiticity of a Hamiltonian ensures the conversation of energy and shapes the physical reality in many systems. When considering non-conservative systems, however, interaction with the environment leads to non-Hermitian dynamics[1–3]. The past decade has witnessed a surge of research on non-Hermitian physics, leading to unprecedented principles, phenomena and applications, as found in open quantum systems[4], electronic systems with interactions[5], classical systems with gain or loss[6–18] and other systems. For instance, by balancing gain and loss under the parity-time symmetry, exceptional points emerge as singular points on the complex energy plane, enabling exotic properties such as unidirectional invisibility[10] and exceptional Fermi arcs[13]. Recently, the concept of non-Hermiticity has been introduced to topological phases of matter, leading to topological physics beyond the Bloch band theory[19–22]. In particular, an intriguing phenomenon is reported, known as the non-Hermitian skin effect[23]. It describes the wave localization toward the open boundaries for an extensive number of bulk modes, which profoundly modifies the band topology and the bulk-boundary correspondence (BBC) and expands the horizon for the study of topological phases of matter, therefore igniting extensive interest[24–38].

Despite the rich physics in non-Hermitian systems, they are primarily studied in zero-dimensional (0D) and one-dimensional (1D) systems. Non-Hermitian physics in higher-dimensions is discussed only very recently in theories[39–41]. Meanwhile, the experimental quest for higher-dimensional non-Hermitian systems is still absent. On another front, the rising of higher-order topological insulators (HOTIs) provides a realm where band topology delicately interplays with crystalline symmetries and leads to multidimensional topological physics[42–58]. At the interface between the non-Hermitian physics and HOTIs, little is known in theories or experiments.

Here, we experimentally demonstrate the higher-order non-Hermitian skin effect in a two-dimensional (2D) acoustic HOTI. The HOTI is realized using two types of coupled resonator acoustic waveguides (CRAWs). The larger waveguides act as site whisper-gallery acoustic resonators, while the smaller waveguides play the role of links and couplers between the site resonators. A biased loss configuration is carefully designed and introduced to the link resonators to realize an anisotropic coupling and hence the non-Hermitian skin effect. When the system has open boundaries along one direction, non-Hermiticity drives wave localizations toward 1D boundaries. When boundaries along both directions are opened, the bulk, edge and corner states all collapse at specific geometric corners. Such a higher-order non-Hermitian skin effect is directly observed from the sound field scanning. Interestingly, our CRAW system supports two degenerate whisper-gallery modes, i.e., the anti-clockwise mode and the clockwise mode, which emulate the spin-up and spin-down states, respectively. We find that the localized states due to non-Hermitian skin effect carry spin momentums. Depending on different spin polarizations, wave localization can happen at different open boundaries and can be modulated upon different loss configurations, exhibiting an interesting spin-dependent and boundary-selective non-Hermitian skin effect.

## Results

### Spinful HOTIs and the higher-order non-Hermitian skin effect.

A unit-cell of our sonic crystal (SC) is shown in Fig. 1a, which consists of four evenly distributed "site" whisper-gallery resonators that are coupled to each other through the "link" resonators. Each resonator has a ring shape comprising alternating air layers and the acoustically rigid material (e.g., the photosensitive resin in our experiments). Such a layered medium

gives rise to an effective refractive index larger than that in air and therefore supports clockwise and anti-clockwise acoustic whispering-gallery modes[59]. In the Hermitian limit, the SC is an acoustic analog of the spinful 2D Su-Schrieffer-Heeger (SSH) model (see Supplementary Information for the derivations of the model). The acoustic band structure is numerically calculated for the 2D SC and presented in Fig. 1b. Because of the degeneracy between the spin-up and spin-down states, each curve in the figure represents two spin-degenerate bands. Considering the band gap between the second and the third bands, different topological properties can be brought forward, from a HOTI, a gapless phase to a trivial insulator, by changing the geometry parameter $\delta$, as shown in Fig. 1b. Specifically, a positive $\delta$ makes the inter-unit-cell couplings stronger than the intra-unit-cell couplings. This leads to a spinful HOTI with a non-trivial bulk dipole polarization $\mathbf{P} = (\frac{1}{2}, \frac{1}{2})$ for each spin-polarization as indicated by the parity inversion between the $\Gamma$ and X points. Such a spinful HOTI hosts gapped edge states and in-gap corner states. Taking into account the spin degeneracy, there are two degenerate edge states at each edge and two degenerate corner states at each corner when the system has open boundaries. In contrast, a negative $\delta$ makes the inter-unit-cell couplings weaker than the intra-unit-cell couplings and hence gives rise to a trivial insulator. At $\delta = 0$, the acoustic bands are gapless, signifying the transition between the HOTI and the trivial insulator.

To realize the higher-order non-Hermitian skin effect, a biased acoustic loss configuration is designed on the link waveguides and is realized by inserting dissipative porous sponge to the air slits, as illustrated by the green regions in Fig. 1a. The dissipation is numerically modeled by a complex sound velocity, $c = (1 + i\gamma) \times 343$ m/s. Such a design introduces anisotropic couplings for the upward (rightward) and downward (leftward) propagating waves, and therefore realizes non-Hermitian skin effect. The non-Hermitian skin effect is often indicated by the significant discrepancy between the energy spectra under the open boundary condition (OBC) and those under the periodic boundary condition (PBC). We calculate the acoustic spectra for our SC subject to the OBC and the PBC. The corresponding topological phase diagrams are extracted and presented in Fig. 1c, which show a clear discrepancy at $\gamma > 0$ (see Supplementary Information for more discussions). This indicates the non-Hermitian skin effect. More concretely, when the system has OBC in the $y$ direction and PBC in the $x$ direction, the non-Hermiticity drives 2D bulk states to be localized toward the 1D open boundaries. We find that the spin-up states, including both the bulk and edge states, are localized toward the lower boundary, whereas the spin-down bulk and edge states are localized toward the upper boundary, as schematically illustrated in Fig. 1d (left panel denotes the Hermitian case while the right panel denotes the non-Hermitian case). This can be understood from the perspectives of the wave dynamics. Specifically, the acoustic loss in the $y$-direction dissipates the energy of an upward spin-up mode, leaving it the only route accumulating downwards, strikingly different from the Hermitian case where both the upward and downward propagations are allowed. Similarly, the spin-down modes can only accumulate upwards. When the system has OBC in the $x$ direction and PBC in the $y$ direction, the spin-up states are localized toward the left boundary, whereas the spin-down states are localized toward the right boundary. These results indicate the spin-polarized non-Hermitian skin effect-a unique feature in our system.

When the boundaries along both the $x$ and $y$ directions are opened (see Fig. 1e), the non-Hermitian skin effect is manifested in multiple dimensions where all the states below and in the band gap, including the 2D bulk, 1D edge and 0D corner states, become

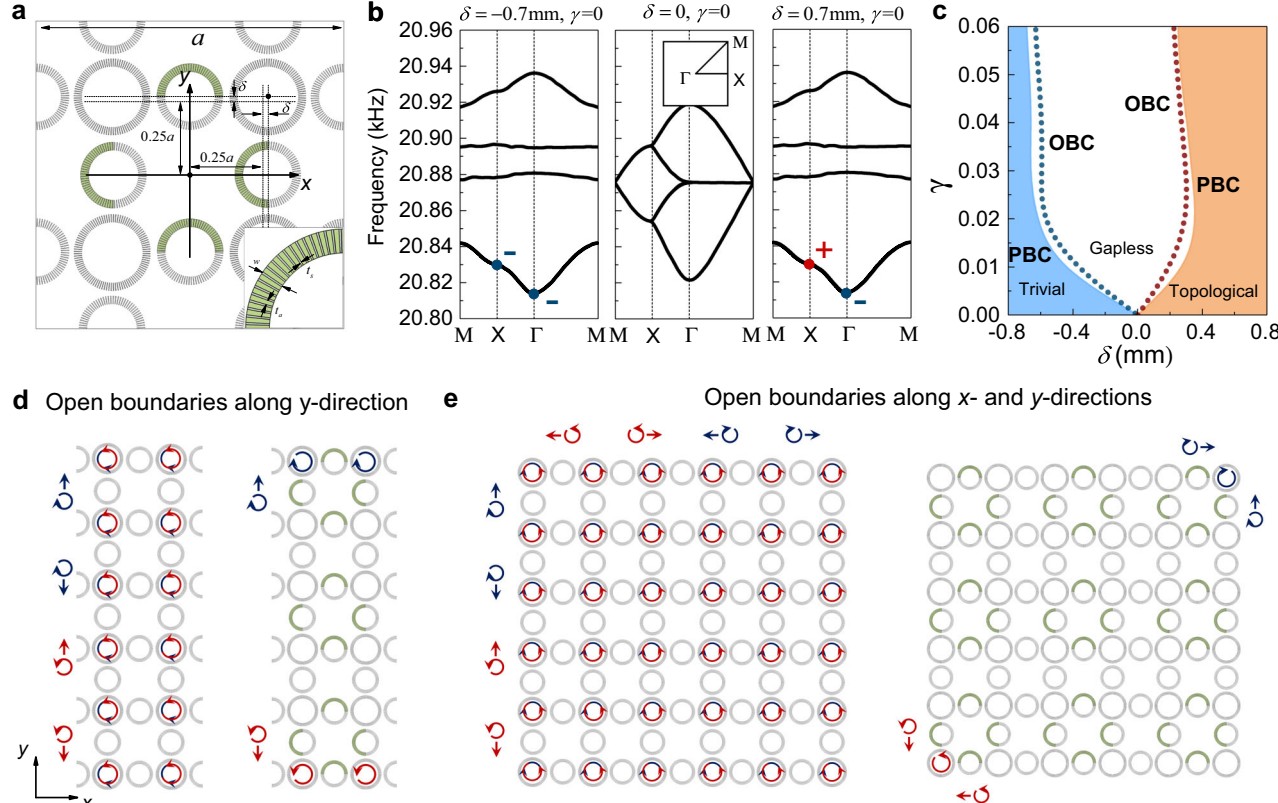

**Fig. 1 The acoustic HOTI and the spin-polarized higher-order non-Hermitian skin effect. a** Unit-cell of the SC, consisting of four site CRAW ring resonators, coupled to each other by smaller link ring resonators. Each site ring comprises 200 alternating layers of rigid material (gray color) and air (the white background) and each link ring contains 160 alternating layers. The geometric parameters are taken as $t_s = 0.4$ mm, $t_a = 1$ mm and $w = 5.7$ mm, with the lattice constant $a = 22$ cm (in the simulations, the SC structure is scaled up by 9.5% to model the geometric errors in the experimental samples). The gap between the site rings and the link rings is quantified by $\delta$, which is tuned to control the intra-cell and inter-cell couplings. Non-Hermiticity is introduced by adding acoustic dissipation in the green regions (realized by porous sponge). **b** Acoustic band structures calculated for the Hermitian SC (i.e., $\gamma = 0$) with $\delta = -0.7$ mm, 0 and 0.7 mm (inset shows the Brillouin zone). A parity inversion at the X point is observed (the lowest band gap is considered), as indicated by the flip of the "+" and "−" signs (respectively representing the s-/d-like states and the p-like states). This is associated with the topological phase transition. **c** Break-down of the conventional BBC in the non-Hermitian cases ($\gamma \neq 0$), signified by the inconsistence of the critical band gap-closing points under the PBC (labeled by the colored shadings) and OBC (labeled by the dotted lines). This indicates the emergence of non-Hermitian skin effect. **d, e** Physical pictures of the spin-polarized higher-order non-Hermitian skin effect. The left panels illustrate the Hermitian cases while the right panels depict the cases with acoustic loss.

skin modes localized toward the corner boundaries. Remarkably, because of the spin-dependent non-Hermitian skin effect, the spin-up states accumulate toward the lower-left corner, whereas the spin-down states accumulate toward the upper-right corner. The emergence of the spin-dependent corner skin modes is the key ingredient in this work.

**Demonstration of non-Hermitian skin effect in one dimension.** We first demonstrate the non-Hermitian skin effect both numerically and experimentally in the system with OBC along the $y$ direction and PBC along the $x$ direction. Figure 2a gives the calculated band structure of a ribbon-like supercell for the Hermitian SC ($\gamma = 0$) with the aforementioned boundary conditions. The figure shows that edge states emerge in the bulk band gap, with four degenerate states at each $k$ (see more details in Supplementary Information). We further present in Fig. 2a the acoustic wavefunctions of two eigen-states, which show the typical features of a bulk state (marked by the gray diamond) and an edge state (marked by the yellow star). We find that the spin-up and spin-down edge states are degenerate and form interference standing-wave patterns (as indicated by the green thick arrows).

When the loss is introduced, the bulk states are no longer extended, but rather become localized around the edge boundaries. The emergence of such edge-like skin modes indicates the non-Hermitian skin effect, which is verified by the calculation results in Fig. 2b with $\gamma = 0.02$. As shown in the figure, the original bulk states are indistinguishable from the edge states and the wavefunctions indicate both of them are localized around the edge boundaries. More significantly, both the skin modes and the edge states demonstrate spin polarizations. The spin-up states are localized at the lower boundary, while the spin-down states are localized at the upper boundary. These wavefunction features, reflecting the spin-dependent non-Hermitian skin effect, are distinct from the Hermitian case.

Experimentally, we measure the extended bulk states and the localized skin modes in two block samples (as schematically illustrated in Fig. 2c, d; see more details in Supplementary Information). To visualize the spectral features, two types of pump-probe measurements are designed. An acoustic plane-wave-like source is generated and guided into the sample through a rectangular waveguide. For the bulk probe, the detector is placed in the bulk region, whereas for the edge probe, the detector is placed in the edge region (see Methods and Supplementary Information for more details on the measurements and error

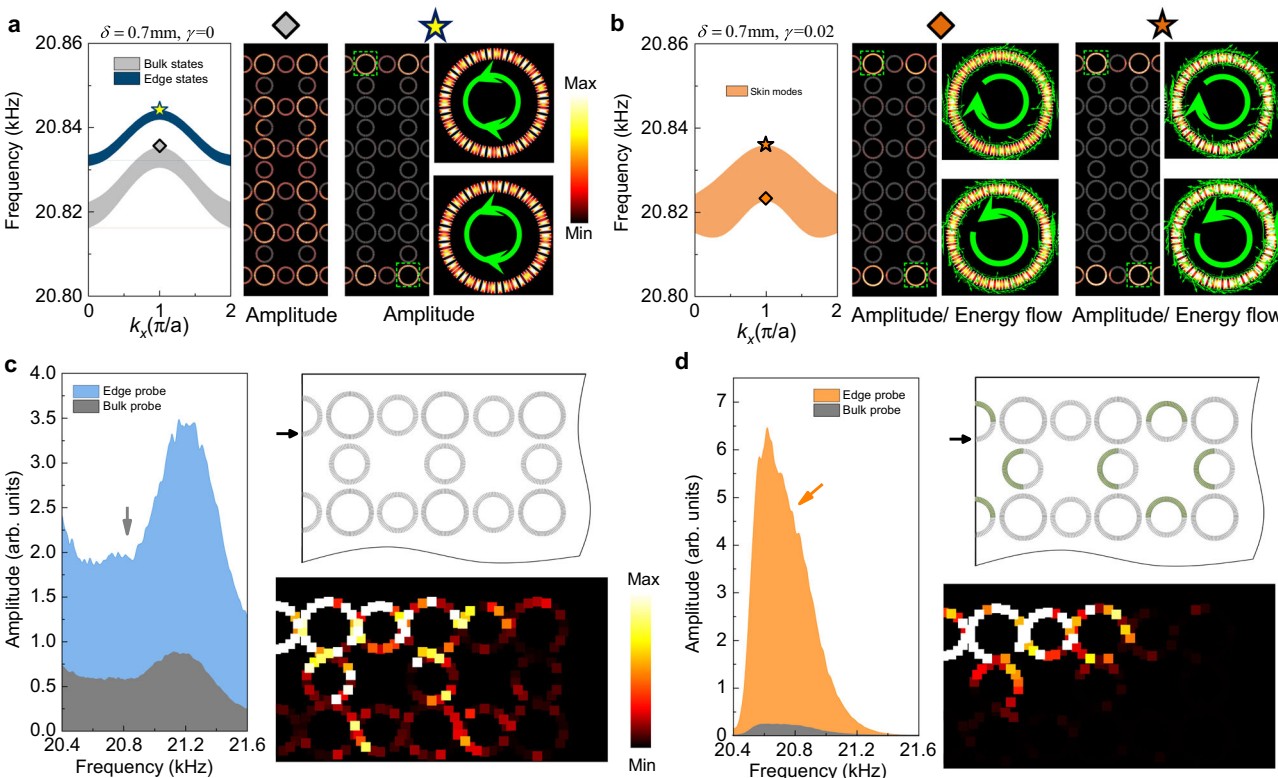

**Fig. 2 Topological edge states and non-Hermitian skin modes. a** Calculated eigen-spectrum for a supercell with PBC along the *x* direction and OBC along the *y* direction in the Hermitian regime. Acoustic pressure field profiles for the two marked states (gray diamond and yellow star) are presented, showing a typical extended bulk state and a topological edge state, whose mode patterns are standing waves due to the spin-degeneracy. **b** The same as **a**, only with acoustic loss. Due to the non-Hermitian skin effect, the original bulk states and topological edge states are no longer distinguishable and become edge-like skin modes, as shown by the acoustic pressure field profiles for the two marked states (orange diamond and star). Interestingly, the non-Hermitian skin effect is spin-polarized, with spin-up states accumulating toward the lower boundary while the spin-down states accumulating toward the upper boundary. This feature is in sharp contrast to the topological edge states in **a**. **c, d** Experimental measurements on the extended bulk states and the non-Hermitian skin modes, respectively. The transmission spectra are measured for the bulk (gray) and edge (blue or orange) pump-probes. The corresponding acoustic field profiles for two excited states (marked by the gray and orange arrows) are also presented, with a piece of SC sample sketched to illustrate the scanning domain (see Supplementary Information for more details), where the black thick arrow indicates the input plane-wave-like signal.

analyses). The detected signals for the cases without and with the dissipation are shown in Fig. 2c, d, respectively. We remark that although the acoustic loss in air is inevitable, it is much weaker than the designed acoustic loss in our samples (which is realized using porous sponge). The dominant effect of the designed loss leads to visible differences between the pump-probe spectra in Fig. 2c, d. Specifically, in Fig. 2c, the bulk-probe signal is salient, whereas, in Fig. 2d, the bulk-probe signal is strongly suppressed. Besides, the spectral shape of the edge probe is also significantly modified. We further scan the acoustic wave fields for two excited states (marked by the arrows in Fig. 2c, d), which illustrate an extended bulk state and the wave localization toward the boundary for the skin mode, agreed well with the spectra. These experimental features clearly demonstrate the non-Hermitian skin effect.

**Demonstration of the higher-order non-Hermitian skin effect in two dimensions.** We now study the system with OBC in both the *x* and *y* directions. In the Hermitian regime, since the system is a spinful HOTI, spin-polarized edge and corner states emerge on the edge and corner boundaries, respectively. We calculate the eigen-states of a box-shaped supercell with $3 \times 3$ unit-cells. The spectrum is shown in Fig. 3a, where the bulk, edge, and corner states can be identified through their wavefunctions, as illustrated by the right-side panels. Because of the spin-degeneracy, each bulk, edge, or corner state is doubly degenerate and

the eigen-state wavefunctions often show the interference patterns between the spin-up and spin-down states. In contrast, in the non-Hermitian case, all the bulk, edge, and corner states become localized around geometric corners. Depending on their spin polarizations, the spin-up modes are localized toward the lower-left corner, while the spin-down modes are localized toward the upper-right corner. Specifically, although we can still identify eight corner modes in the spectrum, all these modes are localized at the upper-right or the lower-left corners. Similarly, the number of the edge and bulk states remains the same, while they also become localized toward the upper-right or the lower-left corners. These anomalous phenomena indicate the higher-order non-Hermitian skin effect. Note that our higher-order non-Hermitian skin effect has different meaning from the second-order non-Hermitian skin effect reported in Ref. [39] where only $\mathcal{O}(L)$ modes are localized at the corners as corner skin modes for the 2D finite system with a size of $L \times L$. In our system, in contrast, all eigenstates become corner skin modes due to the non-Hermitian effects.

We remark that our system has both higher-order topology and non-Hermitian skin effect, which is in a unique regime that has not yet been studied in neither theory nor experiments. The results in this work show that in HOTIs, the non-Hermitian effect can make all the 2D bulk, 1D edge and 0D corner states become non-Hermitian corner skin modes. With our unique design, the non-Hermitian skin effect also becomes pseudospin dependent. It

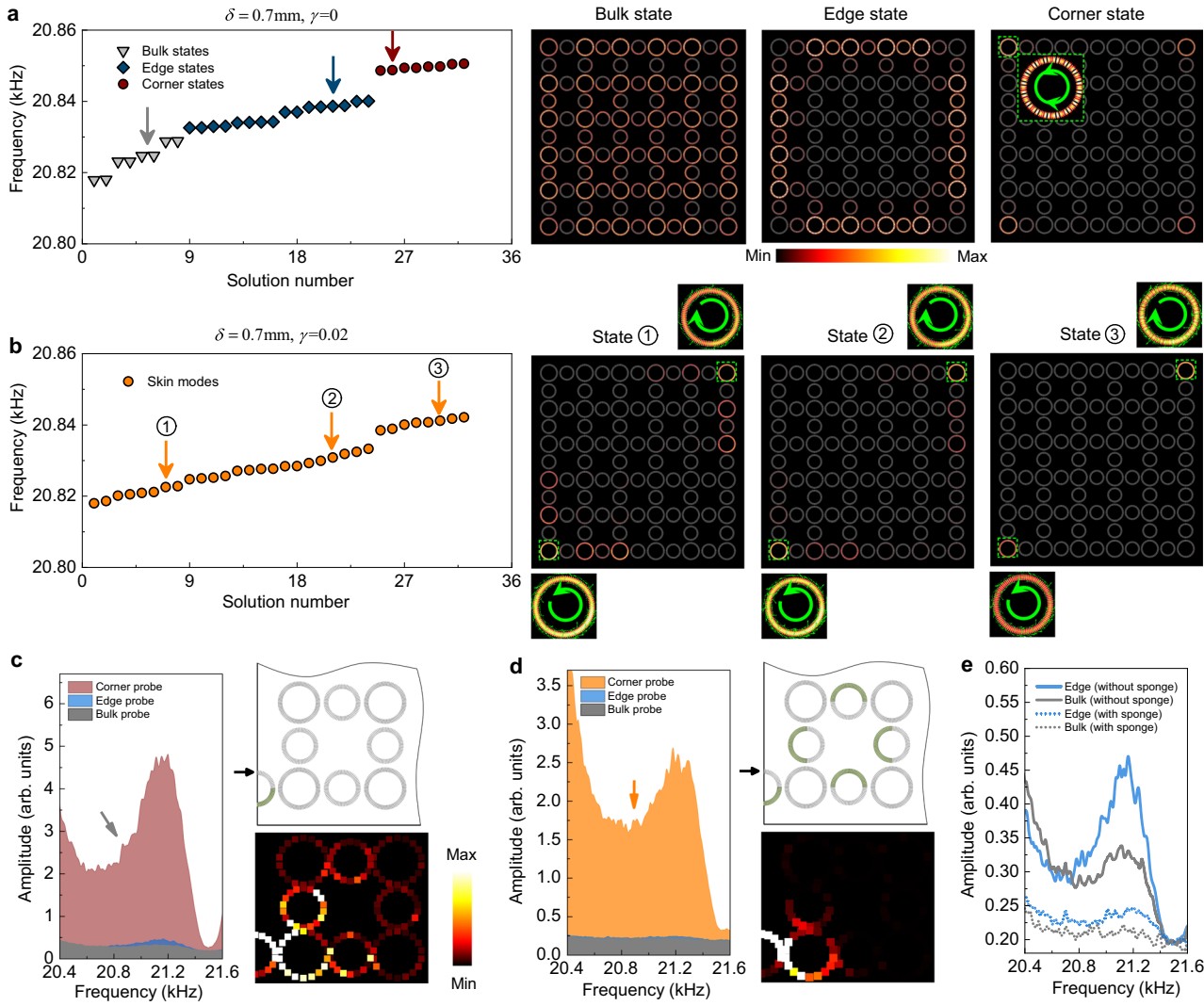

**Fig. 3 Higher-order non-Hermitian skin effect. a** Calculated eigen-spectrum for a box-shaped SC (in the Hermitian regime) with OBC in both the *x* and *y* directions. There are eight corner states (dark red), sixteen edge states (dark blue), and eight bulk states (gray). The corresponding acoustic wavefunctions for the bulk, edge, and corner states (marked by arrows) are presented. **b** The same as **a**, but with acoustic loss. The higher-order non-Hermitian skin effect is manifested by the emergence of spin-polarized, corner-like skin-modes. Depending on the spin-polarizations, the corner-like skin modes reside near the lower-left (for spin-up modes) and upper-right (for spin-down modes) corners. **c–e** Experimental demonstration of the higher-order non-Hermitian skin effect. Both the transmission spectra (for the bulk, edge, and corner pump-probes) and the scanned acoustic pressure profiles are presented, which show consistency with the simulations.

is worth mentioning that higher-order topology is unnecessary for the emergence of the higher-order non-Hermitian skin effect, as shown in refs. [39,40], which is confirmed here by the numerical study on a trivial insulator in the Supplementary Information.

We then measure the spectral properties of the HOTI for samples with OBC in both the *x* and *y* directions. We study two samples, one without the designed acoustic loss and the other with the designed acoustic loss. As shown in Fig. 3c, in the first sample, there are noticeable features of extended waves. In contrast, Fig. 3d indicates that in the second sample, the waves are localized around the corner, demonstrating the higher-order non-Hermitian skin effect. Moreover, compared to the peak signal for the corner probe in the first sample (corresponding to the corner states), in the second sample, the corner probe shows much broadened spectrum, corresponding to the skin modes. Figure 3e indicates that the spectral difference between the bulk-probe and the edge-probe is much weakened in the second sample. This feature is also consistent with the non-Hermitian skin effect which suppresses the difference between the bulk and edge states

in the wavefunctions. These spectral features are further corroborated with the corresponding acoustic wave field scanning for an extended state (Fig. 3c, right panel) and a skin mode (Fig. 3d, right panel).

The robustness of non-Hermitian skin effect in our HOTI is systematically investigated in Supplementary Information, where perturbations on both the geometry and the dissipation are considered. We find that these perturbations impose considerable effects on a small number of the skin modes, making them shifted to different frequencies or become less localized, while the majority of the non-Hermitian skin modes are negligibly affected by the perturbations. Hence, the perturbations only have a limited effect on our system and the experimental features of the higher-order non-Hermitian skin effect observed in this work are robust.

**Versatile wave manipulation based on different configurations of non-Hermiticity.** The above studies indicate that by engineering the loss configuration, intriguing non-Hermitian skin

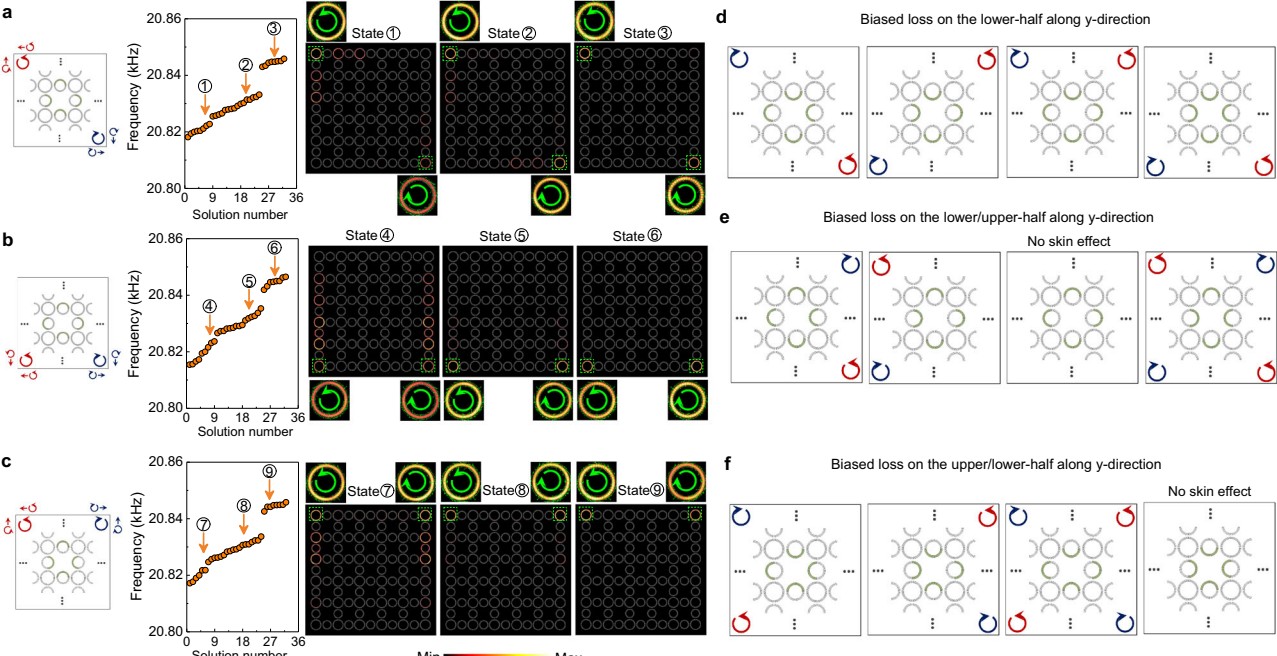

**Fig. 4 Configurable non-Hermitian skin effects. a–c** Calculated eigen-spectra for three box-shaped SCs with different loss configurations (as indicated by the green regions in each unit-cell). Associated with different loss configurations, the higher-order non-Hermitian skin effect is manifested at different corners with different spin-polarizations (as illustrated by the blue and red thick arrows in the box and numerically demonstrated by the acoustic wavefunctions). **d–f** Other schemes to obtain various non-Hermitian skin effects by engineering loss configurations.

effects can be triggered. In fact, versatile non-Hermitian effects can be realized through such an approach. For instance, if we keep the acoustic loss on the upper and lower intra-cell link rings the same as that in Fig. 1a, and re-assign the acoustic loss on the left and right intra-cell link rings to be on the right-half (see Fig. 4a for an illustration), based on a similar analysis as Fig. 1e, the higher-order non-Hermitian skin effect now takes the preference to the upper-left and lower-right corners, hosting the spin-up and spin-down skin modes, respectively. To verify this intuition, the eigen-spectrum of a box-shaped structure, along with the acoustic wavefunctions of three marked eigen-states, are presented in Fig. 4a. Indeed, the expected higher-order non-Hermitian skin effect emerges in the finite sample. When re-assigning the acoustic loss on the left and right intra-cell link rings to be on the opposite-half (see Fig. 4b, c for illustrations), surprisingly, the non-Hermitian skin effect takes preference to the corners in the *x*-direction, i.e., they are either at the lower-left and lower-right corners or at the upper-left and upper-right corners, as verified by the eigen-spectra and acoustic wavefunctions. By using simple permutations and combinations, we further generate twelve selection rules for the non-Hermitian skin effect, in addition to the existing ones, as summarized in Fig. 4d–f, which provide versatile ways to access and control the spin-polarized non-Hermitian skin modes. Specifically, among the twelve loss configurations, two cases support skin modes at all four corners. This feature looks similar to the topological corner states in the Hermitian regime. However, all the skin modes are spin-polarized, which are strikingly different from the Hermitian case. In addition, two cases show that the non-Hermitian skin effect can be disabled when specific loss configurations are designed, despite the existence of non-Hermiticity.

## Discussion

The discovered multidimensional, spin-polarized higher-order non-Hermitian skin effect unveils a fundamental phenomenon in non-Hermitian systems and lays the foundation for the studies of non-Hermitian band topology and wave dynamics in 2D and 3D systems. In these non-Hermitian systems, the synergy between non-Bloch band topology, non-Hermitian skin effect, non-Hermitian parity-time (PT) symmetry, and exceptional singularities (such as the exceptional points, rings, and surfaces) may lead to unprecedented phenomena and potential applications. For instance, the interplay between non-Hermitian effects, non-Bloch bands and the spatial dimension starts to attract theoretical research interest only very recently[39–41]. It is shown that in 2D systems, non-Bloch PT symmetry breaking universally approaches zero as the system size increases. This is fundamentally different from the 1D systems and reveals the interesting interplay between non-Bloch bands and spatial dimensionality[41]. Our study demonstrates that versatile non-Hermitian skin effects can be achieved by engineering the loss configurations, which pave the way for controllable wave dynamics. The design in this work also offers an efficient route towards non-Hermitian systems with spin degrees of freedom and can be generalized to other physical systems, such as on-chip photonics where the coupled resonator waveguides are mature material platforms for integrated technologies.

## Methods

**Experiments**. The present SC consists of blocks made of photosensitive resin (modulus 2765 MPa, density 1.3 g/cm³). We utilize a stereo lithography apparatus to fabricate the samples with a geometric tolerance roughly 0.1 mm. In the experimental measurements, the samples are enclosed in the vertical direction by two acoustically rigid flat plates to form a waveguide. The height of our samples is chosen to be 6 mm, such that around the operating frequency ~21 kHz, only the fundamental waveguide mode is excited. This ensures the validity of the 2D approximation.

The experimental data in Figs. 2 and 3 are measured from the following procedure. We use an acoustic transducer to generate an acoustic white signal, which are guided into the samples from a rectangular waveguide. An acoustic detector (B&K-4939 ¼-inch microphone) is used to probe the excited acoustic pressure field from open channels (with diameter slightly larger than the detector) on the top of the waveguide. For each ring resonator, there are 18 open channels forming a ring-shape to probe the acoustic wave fields. The neighboring channels are separated by ~3 mm and ~1 mm respectively for the site ring and the link ring

(see Supplementary Information for more details). Such small distances make sure the acoustic fields are properly resolved. The data are collected and analyzed by a DAQ card (NI PCI-6251). For the measurements of the acoustic field distributions, the same set-up is used, only with fixed exciting frequency. The acoustic pressure fields of the ring resonators are probed by manually moving the detector. The collected data are post-processed to generate the color maps.

There are several factors that may lead to the discrepancy between the experimental and simulated results. First, the fabrication error is about 0.1 mm and the designed SC has a minimum geometric parameter of 0.4 mm, leading to relatively large geometric errors. In the simulations, we scale-up the SC structure by 9.5% to model the overall geometric errors of the experimental samples. Without the geometric fitting, the frequency shift between the experiments and simulations is roughly 6-8%. Second, the adding of porous sponge might cause the shift of the resonant frequencies of the link rings, which can also lead to discrepancy between the experiments and simulations. Third, the designed HOTI has a band gap width of roughly 40 Hz, yielding the bandgap-to-midgap ratio ~0.2%. Such a small band gap is difficult to be resolved in the experiments, leading to much broadened pump-probe spectra compared to the eigen-spectra in simulations. Nevertheless, due to the existence of an extensive number of skin modes enabled by non-Hermiticity, our experimental results are still able to demonstrate the localization effect associated with the non-Hermitian skin effect (see Supplementary Information for more discussions).

**Simulations**. Numerical simulations in this work are conducted using the 2D acoustic module of a commercial finite-element simulation software (COMSOL MULTIPHYSICS). The resin blocks are treated as acoustically rigid boundaries. The mass density and sound velocity in air are chosen as 1.21 kg/m³ and 343 m/s, respectively (in the cases with designed acoustic loss, a complex sound velocity is taken for the loss domains to characterize the dissipation). To model the geometric errors of the printed samples, in simulations, the SC structure is scaled up by 9.5%. In the eigen-evaluations, the four boundaries of the unit-cell along both the $x$ and $y$ directions are set as Floquet periodic boundaries. The boundaries of the ribbon-like supercells are set as Floquet periodic boundaries along the edge direction, with the perpendicular direction set as plane wave radiation boundaries. The boundaries of the corner samples are set as plane wave radiation boundaries. Note that enough air space should be created between the edge/corner boundaries and the simulation boundaries in order to mimic the physical boundaries in the experiments.

## Data availability
Our data is generated from the commercial finite element software COMSOL under private user license which cannot be made public. The essential data is available in the main text and the Supplementary Information. Additional information is available from the corresponding authors upon reasonable request.

## Code availability
All codes related to the data in the main text and the Supplementary Information are programmed in COMSOL with private user license and therefore are only available from the corresponding authors upon reasonable request.

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

## Acknowledgements
X.J.Z., Y.T., M.H.L. and Y.F.C. are supported by the National Key R&D Program of China (2017YFA0303702, 2018YFA0306200), the National Natural Science Foundation of China (Grant nos. 51902151, 11625418, 11890700 and 51732006), the Natural Science Foundation of Jiangsu Province (Grant no. BK20190284) and the Fundamental Research Funds for the Central Universities (14380165). J.H.J. is supported by the National Natural Science Foundation of China (Grant no. 11675116), the Jiangsu distinguished professor funding and a Project Funded by the Priority Academic Program Development of Jiangsu Higher Education Institutions (PAPD). X.J.Z. thanks Xue-Yi Zhu and Zhi-Kang Lin for useful discussions.

## Author contributions
X.J.Z. and M.H.L. conceived the idea. X.J.Z. performed the numerical simulations. X.J.Z. and J.H.J. performed theoretical analysis. X.J.Z. and Y.T. performed experimental measurements. X.J.Z. and J.H.J. wrote the manuscript. M.H.L. and Y.F.C. guided the research. All the authors contributed to the discussions of the results and the manuscript preparation.

## Competing interests
The authors declare no competing interests.
