## [Peer Review File · Nature Communications]

Observation of higher-order non-Hermitian skin effectREVIEWER COMMENTS

Reviewer #1 (Remarks to the Author):

This work reports the experimental observation of the higher-order non-Hermitian skin effect in an acoustic higher-order topological insulator (HOTI), where the non-Hermiticity (introduced by loss) drives wave localization toward geometric corners for the 2D bulk states, 1D topological edge states, and 0D topological corner states. Interestingly, the designed HOTI carries pseudo-spin degrees of freedom, leading to a spin-polarized non-Hermitian skin effect, i.e., the non-Hermitian skin modes carrying different spins are localized at different open boundaries. The authors fabricated samples and conducted experimental measurements to visualize the skin effect of sound. By configuring loss domains in each unit-cell, the non-Hermitian skin effect can be modulated. Overall, the demonstration of the non-Hermitian skin effect in higher dimensions is of significance and the spin-polarized property for the skin effect is novel. The work is thorough and complete, and the results are convincing and consistent between simulations and experiments. I recommend publication of the manuscript after the minor comments below are addressed.

1. It is known that the non-Hermitian skin effect does not necessarily require the material to be topologically non-trivial. Is the same conclusion true in this system? The authors should clarify this point and specify the similarity and difference between the non-Hermitian skin effect in the topological insulator and the counterpart in the trivial insulator.
2. An important feature of topological insulators is robustness. Hence, discussions on the robustness of the non-Hermitian skin modes should be included.
3. The references are inadequate. Some related theory papers are missing, for example, Phys. Rev. Lett. 124, 056802 (2020) and arXiv:2102.05059. Some results in the latter are consistent with the results presented here, which in fact provide good support to the current work. Additionally, some breakthroughs in the field of non-Hermitian physics should also be included, for example, Science 368, 760-763 (2020) and Science 368, 763-767 (2020), among many others.

Reviewer #3 (Remarks to the Author):

In this submitted manuscript, Zhang et al observed non-Hermitian skin effect in two-dimensional acoustic metamaterials. The celebrated non-Hermitian skin effect is an extremely intriguing and generic phenomenon that underlies the recent rapid progresses of non-Hermitian topology. So far, most theoretical studies of non-Hermitian skin effect have focused on 1D systems, though some have went beyond 1D. As far as I know, all experimental observations of non-Hermitian skin effect have been in 1D.

Thus, the present work is the first experimental observation of non-Hermitian skin effect in higher dimensions, which is definitely an important work in the field of non-Hermitian physics. In particular, recent theoretical studies have suggested that two and three-dimensional non-Hermitian systems display very different behaviors compared to 1D, and therefore the present experimental observation of higher-dimensional non-Hermitian skin effect is a timely contribution to the non-Hermitian physics field. The authors have found new features beyond the 1D non-Hermitian physics. Overall, I would like to recommend this manuscript to Nature Communications.

I have the following comments for further improving this work.

1) The authors should mention in the paper that their "higher-order non-Hermitian skin effect" has somewhat different meaning compared to some other theoretical works. For example, in some papers [e.g. Kawabata et al, Phys. Rev. B 102, 205118 (2020)], second-order non-Hermitian skin effect in two dimension means that $O(L)$ number of eigenstates are localized at the corner, rather than $O(L^2)$ number of eigenstates. The authors should explicitly distinguish their higher-order non-Hermitian skin effect with this different meaning to avoid potential confusion.

2) The author wrote: "When the system has OBC in the x direction and PBC in the y direction, the spin-up states are localized toward the left boundary, whereas the spin-down states are localized toward the right boundary. " This reminds me of the "Z2 non-Hermitian skin effect" [e.g. PRB, Phys. Rev. B. 101, 195147 (2020)]. Could the authors comment on their relation?

3) The following experiments on non-Hermitian skin effect in one dimension is missing: (I) T. Helbig, T. Hofmann, S. Imhof, M. Abdelghany, T. Kiessling, L. W. Molenkamp, C. H. Lee, A. Szameit,

M. Greiter, and R. Thomale, Generalized bulk-boundary correspondence in non-hermitian topoelectrical circuits, Nature Physics 16, 747 (2020). (II) Shuo Liu, Ruiwen Shao, Shaojie Ma, Lei Zhang, Oubo You, Haotian Wu, Yuan Jiang Xiang, Tie Jun Cui, Shuang Zhang, Non-Hermitian Skin Effect in a Non-Hermitian Electrical Circuit, Research, vol. 2021, Article ID 5608038, (2021). In addition, regarding the following remark: "The synergy between non-Bloch band topology, non-Hermitian skin

effect, non-Hermitian parity-time symmetry and exceptional singularities (such as the exceptional points, rings and surfaces) may lead to unprecedented phenomena and potential

applications", It is helpful to note that the non-Bloch PT symmetry has very different features in higher dimensions (Song et al, arXiv:2102.02230).

We thank all reviewers for their valuable comments and suggestions. The manuscript is carefully revised according to these comments and suggestions. We mark the changes in blue in the main text. The relevant works mentioned by the reviewers are added to the reference list.

Reply to the Reviewer #1

Reviewer's Comments

“This work reports the experimental observation of the higher-order non-Hermitian skin effect in an acoustic higher-order topological insulator (HOTI), where the non-Hermiticity (introduced by loss) drives wave localization toward geometric corners for the 2D bulk states, 1D topological edge states, and 0D topological corner states. Interestingly, the designed HOTI carries pseudo-spin degrees of freedom, leading to a spin-polarized non-Hermitian skin effect, i.e., the non-Hermitian skin modes carrying different spins are localized at different open boundaries. The authors fabricated samples and conducted experimental measurements to visualize the skin effect of sound. By configuring loss domains in each unit-cell, the non-Hermitian skin effect can be modulated. Overall, the demonstration of the non-Hermitian skin effect in higher dimensions is of significance and the spin-polarized property for the skin effect is novel. The work is thorough and complete, and the results are convincing and consistent between simulations and experiments. I recommend publication of the manuscript after the minor comments below are addressed.”

Our reply:

We appreciate the reviewer's careful review and positive comments of our work. The manuscript has been revised according to the reviewer's comments.

Reviewer's Comments

“It is known that the non-Hermitian skin effect does not necessarily require the material to be topologically non-trivial. Is the same conclusion true in this system? The authors should clarify this point and specify the similarity and difference between the non-Hermitian skin effect in the topological insulator and the counterpart in the trivial insulator.”

Our reply:

We thank the reviewer for these comments. Indeed, the higher-order non-Hermitian skin effect does not require the system to be topologically non-trivial. This conclusion is also true in our system, as shown by the numerical studies in the revised Supplementary Information. Nevertheless, the non-Hermitian effect in higher-order topological insulators (HOTIs) is also an intriguing topic. One of our purposes in this work is to study such effects on the edge and corner states, in addition to the main purpose to examine the non-Hermitian effect in two-dimensional (2D) lattices with artificial spin degree of freedom. We indeed find several interesting features, including the higher-order non-Hermitian skin effect, the spin-dependent non-Hermitian skin effect and their

controllability as well as the experimental observation that all states, including the bulk, edge and corner states, become non-Hermitian skin modes.

In the revised Supplementary Information, we show numerically that the higher-order non-Hermitian skin effect also exists in the trivial insulator. Meanwhile, the spin-dependent skin effect persists and all the bulk states become skin modes. These calculations indicate that our system is in the regime where the higher-order non-Hermitian skin effect is very strong. It dominates over the higher-order topology and other effects, and makes all eigenstates spin-dependent skin modes. In the revised Supplementary Information, we include the simulation results for the non-Hermitian trivial insulator and the detailed discussions in Section 6. Correspondingly, in the main text, we clarify clearly that the higher-order non-Hermitian skin effect also exists in topologically trivial systems in Page 8.

Reviewer's Comments

“An important feature of topological insulators is robustness. Hence, discussions on the robustness of the non-Hermitian skin modes should be included.”

Our reply:

We thank the reviewer for the suggestion. The topological edge and corner states in our HOTI are protected by the crystalline symmetry, similar as that have been reported in other SSH-like lattices (e.g., see Ref. [Phys. Rev. Lett. 122, 233903 (2019)]). However, the non-Hermitian skin modes emerge due to the interference effect of the clockwise (anti-clockwise) modes driven by the non-Hermiticity, as schematically illustrated in Fig. 1d-e in the main text. They are not protected by the crystalline symmetry. Inducing perturbations indeed affects the skin modes to certain extent.

To provide more evidence, we conduct systematical simulations to investigate the effect of perturbations of geometries and lossy factors on the skin modes. Three types of geometric perturbations are considered, including the perturbations on geometric corners, edges and bulks. We find that the skin modes are more sensitive to the perturbations on geometric corners and edges, compared with the bulk perturbations. This is due to the fact that the non-Hermiticity drives the resonant modes accumulating towards the edges and then towards the corners. If the edges and corners are perturbed, the interference process is sabotaged, accordingly affecting the formation of the skin modes. On the other hand, the resonant modes in the bulk can always re-route when encountering with perturbations, which therefore impose relatively small effect on the skin modes.

When the designed loss domains are perturbed, the skin modes can also be affected, depending on the perturbation strength. We find that as long as the system is under non-Hermitian condition, the skin modes remain negligibly affected. However, if the perturbations on loss factor lead to break-down of non-Hermiticity, some of the skin modes can be severely disturbed. This demonstrates the important role of non-Hermiticity in the formation of skin modes.

We have included the simulation results on the robustness study for the non-Hermitian skin modes in the revised Supplementary Information as Section 5 and added more discussions in the main text in Page 9.

Reviewer's Comments

“The references are inadequate. Some related theory papers are missing, for example, Phys. Rev. Lett. 124, 056802 (2020) and arXiv:2102.05059. Some results in the latter are consistent with the results presented here, which in fact provide good support to the current work. Additionally, some breakthroughs in the field of non-Hermitian physics should also be included, for example, Science 368, 760-763 (2020) and Science 368, 763-767 (2020), among many others.”

Our reply:

We thank the reviewer for pointing out this issue. In the revised manuscript, we have added more references on the study of the non-Hermitian skin effect and the breakthroughs in the field of non-Hermitian physics. The added references have indices of Refs. [16-18], [27-30], [35], [38] and [40, 41].

Reply to the Reviewer #3

Reviewer's Comments

“In this submitted manuscript, Zhang et al observed non-Hermitian skin effect in two-dimensional acoustic metamaterials. The celebrated non-Hermitian skin effect is an extremely intriguing and generic phenomenon that underlies the recent rapid progresses of non-Hermitian topology. So far, most theoretical studies of non-Hermitian skin effect have focused on 1D systems, though some have went beyond 1D. As far as I know, all experimental observations of non-Hermitian skin effect have been in 1D. Thus, the present work is the first experimental observation of non-Hermitian skin effect in higher dimensions, which is definitely an important work in the field of non-Hermitian physics. In particular, recent theoretical studies have suggested that two and three-dimensional non-Hermitian systems display very different behaviors compared to 1D, and therefore the present experimental observation of higher-dimensional non-Hermitian skin effect is a timely contribution to the non-Hermitian physics field. The authors have found new features beyond the 1D non-Hermitian physics. Overall, I would like to recommend this manuscript to Nature Communications.”

Our reply:

We appreciate the reviewer's careful review and positive comments. The manuscript has been revised according to the reviewer's comments.

Reviewer's Comments

“The authors should mention in the paper that their “higher-order non-Hermitian skin effect” has somewhat different meaning compared to some other theoretical works. For example, in some papers [e.g. Kawabata et al, Phys. Rev. B 102, 205118 (2020)], second-order non-Hermitian skin effect in two dimension means that $O(L)$ number of eigenstates are localized at the corner, rather than $O(L^2)$ number of eigenstates. The authors should explicitly distinguish their higher-order non-Hermitian skin effect with this different meaning to avoid potential confusion.”

Our reply:

We thank the reviewer for being very meticulous. Indeed, the higher-order non-Hermitian skin effect in our system is different from the second-order skin effect that has been reported in Ref. [Kawabata et al, Phys. Rev. B 102, 205118 (2020)] where $O(L^2)$ modes are delocalized through the bulk while $O(L)$ modes are localized at the corners as corner skin modes (here, $L \times L$ denotes the size of the 2D open system). In our system, however, all the states including the bulk, edge and corner states at Hermitian limit become corner skin modes driven by non-Hermiticity. In addition, our system supports pseudospin degrees of freedom and therefore the skin modes are doubled compared with spinless cases. Particularly, the corner skin modes in our system are pseudospin-polarized with pseudospin-up modes localized at the lower-left corner and pseudospin-down modes appearing at upper-right corner. Such a pseudospin-polarization can also be modulated by re-configuring the loss domains (see Fig. 4 of the main text for more details). To avoid possible

confusion, in the revised manuscript, we have included discussions on the difference between our system and that reported in Ref. [Kawabata et al, Phys. Rev. B 102, 205118 (2020)] in Page 8.

Reviewer's Comments

"The author wrote: "When the system has OBC in the x direction and PBC in the y direction, the spin-up states are localized toward the left boundary, whereas the spin-down states are localized toward the right boundary. " This reminds me of the "Z₂ non-Hermitian skin effect" [e.g. PRB, Phys. Rev. B. 101, 195147 (2020)]. Could the authors comment on their relation?"

Our reply:

We thank the reviewer for raising this point. Ref. [Phys. Rev. B. 101, 195147 (2020)] demonstrated the break-down of standard non-Bloch band theory in the symplectic class (in the paper, it refers to the time-reversal symmetry that gives rise to the Kramers degeneracy and ensures the Z₂ topological phase), leading to a Z₂ non-Hermitian skin effect protected by reciprocity. Our system, on the other hand, is more like two copies of decoupled pseudospin-up and pseudospin-down SSH models. For each spinful model, the special design of the loss domains leads to the break-down of reciprocity (see Fig. 1a of the main text), similar to that happening in the prototypical examples for non-Hermitian skin effect (e.g., see Ref. [Phys. Rev. Lett. 121, 086803 (2018)]). This suggests that the standard non-Bloch band theories (e.g., Refs. [Phys. Rev. Lett. 121, 086803 (2018)], [Phys. Rev. Lett. 123, 066404 (2019)], [Phys. Rev. Lett. 124, 056802 (2020)] and [Phys. Rev. Lett. 125, 226402 (2020)]) are applicable to our system, which can help to explain the non-Hermitian skin effect in 1D structures with OBC in one direction and PBC in the other direction. For example, if we have an open structure with OBC in the x direction and PBC in the y direction, for spin-up states, non-Hermiticity drives wave localization toward the left boundary while the spin-down skin modes appear around the right boundary. This phenomenon, interestingly, exhibits the similar properties to the Z₂ non-Hermitian skin effect protected by reciprocity in Ref. [Phys. Rev. B. 101, 195147 (2020)], where a pair of loops of the generalized Brillouin zone describes the localized modes (i.e., the skin modes) at the right and left edges.

Reviewer's Comments

"The following experiments on non-Hermitian skin effect in one dimension is missing: (I) T. Helbig, T. Hofmann, S. Imhof, M. Abdelghany, T. Kiessling, L. W. Molenkamp, C. H. Lee, A. Szameit, M. Greiter, and R. Thomale, Generalized bulk-boundary correspondence in non-hermitian topoelectrical circuits, Nature Physics 16, 747 (2020). (II) Shuo Liu, Ruiwen Shao, Shaojie Ma, Lei Zhang, Oubo You, Haotian Wu, Yuan Jiang Xiang, Tie Jun Cui, Shuang Zhang, Non-Hermitian Skin Effect in a Non-Hermitian Electrical Circuit, Research, vol. 2021, Article ID 5608038, (2021). In addition, regarding the following remark: "The synergy between non-Bloch band topology, non-Hermitian skin effect, non-Hermitian parity-time symmetry and exceptional singularities (such as the exceptional points, rings and surfaces) may lead to unprecedented phenomena and

potential applications", It is helpful to note that the non-Bloch PT symmetry has very different features in higher dimensions (Song et al, arXiv:2102.02230)."

Our reply:

We thank the reviewer for pointing out these nice papers, which have been included in the revised manuscript as Refs. [35] and [38]. In addition, we have added more references on the studies of the non-Hermitian skin effect and the breakthroughs in the field of non-Hermitian physics as Refs. [16-18], [27-30] and [40, 41]. Regarding the reviewer's constructive suggestions on the conclusion remarks, we have added more notes on the non-Bloch PT symmetry (especially paying attention to the results reported in the referred arXiv preprint).

REVIEWERS' COMMENTS

Reviewer #1 (Remarks to the Author):

The revised manuscript has well addressed my comments in the previous round, especially on the non-Hermitian skin effect and topologically associated robustness. In this regard, I would like to recommend its publication as is.

Reviewer #3 (Remarks to the Author):

I have read the revised manuscript and the responses from the authors. I believe that all the raised issues have been satisfactorily addressed, and the revised manuscript can be published as is. The non-Hermitian skin effect is a vital phenomenon in the recent vigorous developments of non-Hermitian physics; its observation beyond one spatial dimension, as done here, is definitely a very significant experimental progress in the field.

Reviewer #1

Comments:

The revised manuscript has well addressed my comments in the previous round, especially on the non-Hermitian skin effect and topologically associated robustness. In this regard, I would like to recommend its publication as is.

Our reply:

We thank the reviewer for his/her time and effort dedicated to our manuscript. The valuable comments and constructive suggestions have helped us to greatly improve the quality of our work.

Reviewer #3

Reviewer's Comments

I have read the revised manuscript and the responses from the authors. I believe that all the raised issues have been satisfactorily addressed, and the revised manuscript can be published as is. The non-Hermitian skin effect is a vital phenomenon in the recent vigorous developments of non-Hermitian physics; its observation beyond one spatial dimension, as done here, is definitely a very significant experimental progress in the field.

Our reply:

We thank the reviewer for his/her time and effort dedicated to our manuscript. The valuable comments and constructive suggestions have helped us to greatly improve the quality of our work.